# Common inflammatory proteins linking frailty and area-level deprivation as key drivers of cardiovascular risk in women
Yu Lin[1,6], Panayiotis Louca[1,6], Ruth C. E. Bowyer[1], Afroditi Kouraki[2], Niccolò Rossi[1], Mary Ni Lochlainn[1], Anthony Kelly[2], Vasileios Georgopoulos[2], Frances M. K. Williams[1], Claire J. Steves [1], Mario Falchi [1,7], Ana M. Valdes [2,3,7] ✉ & Cristina Menni [1,4,5,7] ✉

## Abstract

**Background** Chronic inflammation is linked to frailty and deprivation, both of which are comorbid with cardiovascular diseases (CVD). This study aims to identify inflammatory proteins associated with both socioeconomic deprivation and frailty, and assess their role in mediating cardiovascular risk in a large cohort with independent replication.
**Methods** We included 2144 TwinsUK females aged 37–84 with concurrent measures of frailty (frailty index), index of multiple deprivation (IMD), cardiovascular risk (ASCVD score), and 74 proteins (Olink inflammation panel). A random forest model with SHapley Additive exPlanations identified shared proteomic markers of frailty and deprivation. Linear mixed models assessed associations between selected proteins, IMD, frailty, and ASCVD score. Findings were validated in 57 females from the Nottingham Osteoarthritis study. Mixed-effects Cox regression evaluated associations with 10-year ischemic heart disease risk, and mediation analysis assessed the role of proteins in linking IMD and frailty to ASCVD risk.
**Results** We identify ten pro-inflammatory proteins associated with both frailty and area-level social deprivation. Four of those (TNFSF14, HGF, CDCP1, and CCL11) are consistently positively correlated with ASCVD score in both two cohorts. CDCP1 is also associated with higher incident ischemic heart disease risk (HR [95%CI] = 1.82 [1.17, 2.85]). TNFSF14, HGF, and CDCP1 mediate the association between IMD and ASCVD, as well as between frailty and ASCVD.
**Conclusions** Our findings indicate that inflammatory proteins involved in cellular signalling, growth, and migration are associated with frailty, socioeconomic deprivation, and CVD risk, suggesting that these pathways mediate the impact of socioeconomic deprivation and ageing on CVD risk.

## Plain language summary

Individuals living in deprived areas and those experiencing frailty have a higher risk of cardiovascular disease (CVD), but the biological reasons remain unclear. This study explored whether inflammation, a key process in ageing and disease, could explain this link. We analysed data from over 2000 adult women and identified ten inflammatory proteins linked to both frailty and social disadvantage. Among these, TNFSF14, HGF, CDCP1, and CCL11 were also associated with increased cardiovascular risk, and CDCP1 was further linked to future heart disease events. These findings suggest that these inflammatory proteins may be the missing link that explains how social deprivation and frailty are connected to a higher risk of heart disease. Understanding how this works could help guide more targeted strategies to prevent CVD, especially in vulnerable populations.

Inflammation has been shown to be a causative factor in the incidence of various forms of cardiovascular disease (CVD)[1,2]. CVD is highly prevalent among ageing individuals, particularly those with frailty, and its incidence is elevated in those experiencing socioeconomic deprivation[3,4]. Frailty is an age-related clinical condition, characterised by a deterioration in physiological capacity of organ systems, leading to increased susceptibility to

stressors[5]. Understanding the physiological mechanisms that contribute to the elevated CVD risk in frail and socioeconomically disadvantaged populations is essential for developing realistic, effective interventions[6].

Individuals living with frailty have increased inflammation compared to non-frail individuals[7], a feature of the ageing process, known as 'inflammaging'[8,9]. A large systematic review and meta-analysis,

[1]Department of Twin Research & Genetic Epidemiology, King's College London, London, UK. [2]Academic Rheumatology Clinical Sciences Building, Nottingham City Hospital, University of Nottingham, Nottingham, UK. [3]Nottingham NIHR Biomedical Research Centre, University of Nottingham, Nottingham, UK. [4]Department of Pathophysiology and Transplantation, Università Degli Studi di Milano, Milan, Italy. [5]Fondazione IRCCS Cà Granda Ospedale Maggiore Policlinico, Angelo Bianchi Bonomi Hemophilia and Thrombosis Center, Milan, Italy. [6]These authors contributed equally: Yu Lin, Panayiotis Louca. [7]These authors jointly supervised this work: Mario Falchi, Ana M Valdes, Cristina Menni. ✉e-mail: ana.valdes@nottingham.ac.uk; cristina.menni@kcl.ac.uk

 1

encompassing 43 studies and more than 110,000 participants, confirms that lower socioeconomic status (SES) is significantly associated with elevated levels of systemic inflammatory markers, including IL-6 and CRP[10]. Pro-inflammatory pathways are hypothesised to be a key physiological mechanism that translates socioeconomic health inequalities and mediates the effects of deprivation on higher disease risk, including frailty.

Frailty is also strongly linked to a higher risk of CVD[11-13]. The risk of cardiovascular-related mortality among low SES individuals is 1.6–2.1 times higher compared to those with high SES[14]. Moreover, individuals who develop frailty in late life often exhibit behavioural and cardiometabolic risk factors, which are more common among those with low SES, contributing to an increased risk of CVD[15]. Similarly, osteoarthritis (OA) has also been widely reported to co-exist with CVD, a relationship that cannot be fully explained by ageing, obesity, or lifestyle factors, which may involve complex physiological mechanisms[16]. The coexistence of CVD and other comorbidities among socially disadvantaged individuals suggests the possibility of common risk factors, yet the underlying common inflammatory pathways remain unclear.

In this study, we aim to identify common inflammatory proteins (Olink inflammatory panel) linked to both socioeconomic deprivation and frailty and understand their mediating effect on cardiovascular risk in a large population-based cohort with independent replication. Identifying such proteins may enable early risk stratification, guide personalised interventions, and highlight therapeutic targets. Clinically, these biomarkers could inform future tools to identify frail or socioeconomically disadvantaged individuals at elevated CVD risk due to heightened inflammation and accelerated aging. In this large population-based study, we find ten inflammation-related proteins associated with both frailty and social deprivation. Four of those proteins, TNFSF14, HGF, CDCP1, and CCL11 are also linked to increased cardiovascular risk across both discovery and validation cohorts. Notably, CDCP1 consistently associates with future ischaemic heart disease, suggesting a potential biological pathway connecting social disadvantage, inflammation, and cardiovascular health.

## Methods
### Study population
**Discovery cohort**. Study participants were individuals enroled in the TwinsUK Registry, a national register of adult twins recruited as volunteers without selecting for any particular disease or trait[17]. It is among the most detailed omics and phenotypic bioresource worldwide, including over 14,000 twins comparable to the general population for health and lifestyle characteristics[17]. Historically, the cohort is predominantly female. All twin participants provided informed written consent. The TwinsUK study, approved by the St Thomas' Hospital Research Ethics Committee (REC reference: EC04/015), encompasses longitudinal clinical and multi-omic phenotyping, including substudies such as ours that utilise existing data collected under this protocol[17].

Our study included 2144 twins with concurrent measures of frailty, social deprivation level, cardiovascular phenotypes and inflammation-related markers. Since frailty may change with age, we selected individuals with a maximum gap of 5 years between their Frailty index (FI) and Olink data collection. Since the majority of subjects from TwinsUK exhibited similar socioeconomic status over time, we matched these individuals with IMD data from the closest date to their Olink data collection. This selected cohort comprised the final population for this study.

**Frailty**. Frailty was estimated using the Rockwood FI, a comprehensive measure of the individual's health and functional status. 39 binary outcomes related to health deficit, including physical and mental health, were identified from questionnaires and clinical visits[18]. FI was calculated as the ratio of the individual number of health deficits over the number of completed domains. Individuals were categorised into non-frail ($\leq 0.08$), pre-frail ($> 0.08$) and frail ($\geq 0.25$) according to FI[19]. When used as a continuous trait, FI was log-transformed and standardised.

**SES measurement (area level based)**. The index of multiple deprivation (IMD), a measure of neighbourhood-level socioeconomic status, was determined by the residential address postcode[20]. The IMD score is divided into ten deciles, with the first decile representing the most deprived areas, and the highest decile the least deprived areas, respectively.

Individuals with IMD in the 1st quintile and FI $\geq 0.25$ were defined to be frail and living in a socioeconomic deprived area, while individuals with IMD in the 5th quintile and FI $\leq 0.08$ were defined to be non-frail and living in the least socioeconomically deprived areas.

**Atherosclerotic cardiovascular disease**. The Atherosclerotic Cardiovascular Disease (ASCVD) risk score estimates the individual 10-year CVD risk based on ethnicity, age, sex, and traditional cardiometabolic risk factors (type 2 diabetes, smoking, total cholesterol, HDL cholesterol, systolic blood pressure, and treatment of hypertension) and was calculated as previously described[21]. The ASCVD scores were square root (sqrt)-transformed to reduce skewness in the data distribution and standardised for use in calculating the association within the TwinsUK and Nottingham OA cohorts.

**Lifestyle factors**. Physical activity was measured using the International Physical Activity Questionnaire[22]. Smoking and drinking status were assessed from questionnaires that measured the frequency of alcohol and cigarette consumption. Individuals were identified as current smokers or not according to their smoking habits. Participants with different responses for their drinking habits were classified as never drinking, occasionally drinking, or weekly drinking.

**Dietary information**. A validated 131-item semi-quantitative food frequency questionnaire (EPIC-FFQ)[23] was used to estimate habitual dietary information. From FFQs, food items, macro- and micronutrient intakes were determined using FETA software[23,24]. From intakes, we also calculated indexes to represent the whole dietary pattern, including the healthy eating index (HEI), which characterises intakes of foods and nutrients and it is understood to be associated with chronic diseases[25]

**Ischaemic heart disease**. Incident ischaemic heart disease was ascertained through self-reported questionnaires administered to participants at regular follow-up intervals. Specifically, participants were asked whether a doctor had formally diagnosed them with ischaemic heart disease since their previous assessment. We defined incident cases as those reporting a diagnosis of ischaemic heart disease during the follow-up period who had not reported such a diagnosis at baseline.

**Replication cohort**. Results were replicated in 57 female individuals ($\leq 79$ years) with knee OA from the Nottingham OA study[26,27], with measurement of protein profiles from the Olink panel and ASCVD-related risk factors. Systolic blood pressure (SBP) was imputed based on age and BMI. The imputation of SBP for the Nottingham OA cohort was conducted using a linear regression model derived from data on age and BMI from patients with knee OA in the INSPIRE cohort. The INSPIRE Study, which investigated the effects of dietary fibre and exercise on knee OA, recruited patients from the Nottingham area between May 2022 and December 2023. The participants of the Nottingham OA cohort and the INSPIRE cohort came from the same geographic area, ensuring that those individuals shared similar ethnic backgrounds and lifestyles. The SBP model was developed using three variable combinations: age only, age and BMI, and age, BMI, and sex. The model utilising age and BMI, which had the lowest Akaike Information Criterion and Bayesian Information Criterion, was selected to predict SBP in the Nottingham OA cohort. Ethical approval for the Nottingham OA study was obtained from the East Midlands – Derby NHS Research Ethics Committee (20/EM/0065, 18/EM/0154, respectively) and the Health Research Authority (protocol no: 19098, 18021). All participants provided informed written consent.

**Olink panel profiling.** 74 inflammation-related proteins from plasma samples were profiled using the Olink Proximity Extension Assay (PEA) technique (v.3021 panels), as previously described[28–30]. Briefly, PEA uses a pair of antibodies labelled with unique complementary oligonucleotides (proximity probes) to bind to their specific target protein in a sample. This binding brings the probes into proximity, allowing them to hybridise and enabling DNA amplification of the protein signal, which is then quantified using next-generation sequencing. Plasma samples were randomly allocated to a 96-well plate including six Olink controls and 9 wells populated with a master plasma mix distributed across all 13 plates. The protein levels in the samples were reported using Normalised Protein Expression (NPX), a relative quantification unit on the $\log_2$ scale. Plasma samples from the TwinsUK cohort were measured in two batches at different time points. We excluded proteins that were measured in only one batch of Olink data, and imputed missing data using a KNN-based imputation method. This process resulted in the selection of 69 out of 74 inflammation-related proteins for downstream analysis method.

**Statistics and reproducibility.** Statistical analysis was performed using R 4.2.0 and the machine learning models were constructed using Python 3.7.0.

**Frailty and IMD association.** Linear mixed models were used to explore the relationship between the FI and IMD, adjusting for (i) age, BMI and family relatedness (random effect); (ii) further adjusting for age stopping full-time education, drinking, smoking, physical activity and HEI. The 95% confidence intervals for effect sizes were calculated using the R function confint(), which provides more estimates using profile likelihood-based confidence intervals for fixed effects.

We used a two-step approach to identify proteins associated with deprivation and frailty. We first employed a Random Forest model with SHapley Additive exPlanations (SHAP)[31] to identify the 20 most predictive proteins (based on feature importance) for distinguishing frail individuals in deprived areas from non-frail individuals in advantaged areas. We then tested these proteins individually using linear mixed models, with deprivation and frailty as separate outcomes, adjusting for age, BMI, batch, and family structure.

To identify the most important features, a random forest model with SHAP approach was used to classify socially deprived frail VS non-deprived & non-frail conditions. Age-matched individuals living in the least deprived area without frailty were selected as controls for subjects living in deprived area and with frailty. Individuals were split into a training set and validation set (70%/30%) using a group split method for the feature selection. Individuals from the same family were always allocated as a group to avoid separating twins into different sets[31]. In addition to the Random Forest model, a Partial Least Squares Discriminant Analysis (PLS-DA) was developed using the identified key features. Performance was tested using the area under the receiver operating characteristic (AUC) after five-fold cross-validation.

We selected the top 20 important proteins from SHAP analysis following a widely used rule of thumb, aligning with several other omics studies employing SHAP for feature ranking[32–34]. This approach balances interpretability and biological relevance while avoiding overfitting. We then explored the relationship between the top 20 inflammation-related markers from SHAP and (i) frailty (as continuous trait) and (ii) social deprivation (IMD) using linear mixed models, adjusting for age, BMI, batch effects, family relatedness and multiple testing (Benjamini–Hochberg). Identified proteins significantly associated with both FI and social deprivation ($P < 0.05$ and False Discovery Rate [FDR] <0.1) were taken for downstream analysis.

**Investigating the role of traditional, environmental and lifestyle factors on the association of the identified proteins.** Linear mixed models adjusting for age, BMI, batch and family relatedness (as random

effects) and multiple testing (Benjamini–Hochberg), were used to investigate the association between the identified protein markers and (i) educational attainment; (ii) physical activity; (iii) HEI; (iv) the ASCVD score. Protein markers with an FDR < 0.1 were considered statistically significant.

**Replication of protein signatures in the Nottingham OA study.** Linear regression models adjusting for age, BMI and batch were used to determine the association between the identified inflammation-related markers and ASCVD risk score in the female participants from the Nottingham OA study, adjusting for age and BMI and multiple testing (Benjamini–Hochberg).

**Association between the ASCVD-associated proteins and incident ischaemic heart disease in TwinsUK.** For the survival analysis, the time-to-event outcome was incident ischaemic heart disease. Cases were defined as participants diagnosed with ischaemic heart disease within 10 years following the collection of Olink inflammatory data. Individuals with a prior history of ischaemic heart disease or who developed the disease during follow-up were excluded from the control group. Age-matched controls were selected at baseline for subjects with ischaemic heart disease. The Kaplan-Meier method was used to estimate the probability of ischaemic heart disease for individuals in the top and bottom tertiles of four replicated protein markers. Log-rank tests were performed to assess whether there was a statistically significant difference in disease risk between groups. To evaluate the impact of other covariates, including BMI and other CVD-related factors (smoking status, diabetes diagnosis, use of anti-hypertensive medication, total cholesterol level, HDL cholesterol level), on ischaemic heart disease risk, we used mixed-effects Cox regression models to estimate the association between externally validated markers and 10-year ischaemic heart disease risk, accounting for batch effects, family relatedness, and potential confounders.

**Exploring the mediatory role of the replicated proteins in TwinsUK in the association between IMD, frailty and ASCVD.** We ran a mediation analysis to investigate the mediatory role of the identified proteins in the association between (i) IMD and ASCVD risk; (ii) FI and ASCVD risk. We conducted causal mediation analyses following the Baron and Kenny framework[35]. First, we evaluated the three essential mediation assumptions: (1) significant association between the independent variable and the dependent variable, (2) significant association between the independent variable and the mediator, and (3) significant association between the mediator and the dependent variable when controlling for the independent variable. After confirming these assumptions, we implemented formal causal mediation analysis using the 'mediate' function from the R package 'mediation' (version 4.5.0)[36]. Each mediator was analysed independently. We determined significant mediation based on both statistical significance ($p < 0.05$) and the magnitude of the indirect effect. The variance accounted for (VAF) was determined as the ratio of the indirect-to-total effect and distinguished the proportion of the variance explained by the mediation process (the proportion of the effect of social deprivation and frailty on CVD that goes through the protein markers).

### Reporting summary

Further information on research design is available in the Nature Portfolio Reporting Summary linked to this article.

## Results

An overview of our study design is displayed in Fig. 1. We included 2144 females from TwinsUK[17], and 57 middle-aged to older female individuals with knee OA from the Nottingham OA study[26,27], with protein data from the Olink Target 96 Inflammation panel. This panel uses PEA technology to sensitively quantify inflammation-related proteins, including cytokines,

**Fig. 1 | Overview of the study design.** A total of 2144 TwinsUK participants were analysed to explore biological links between social deprivation, frailty, and cardiovascular risk. Random Forest with SHapley Additive explanation (SHAP) identified the top 20 proteomic features distinguishing the most deprived frail and least deprived non-frail individuals. These proteins were examined for associations with the IMD, frailty, and ASCVD risk in TwinsUK, and validated in an independent replication cohort. Associations of replicated proteins with 10-year ischaemic heart disease risk were assessed in age-matched TwinsUK participants. Mediation analyses were performed to examine whether shared inflammatory proteins linked deprivation, frailty, and cardiovascular risk.

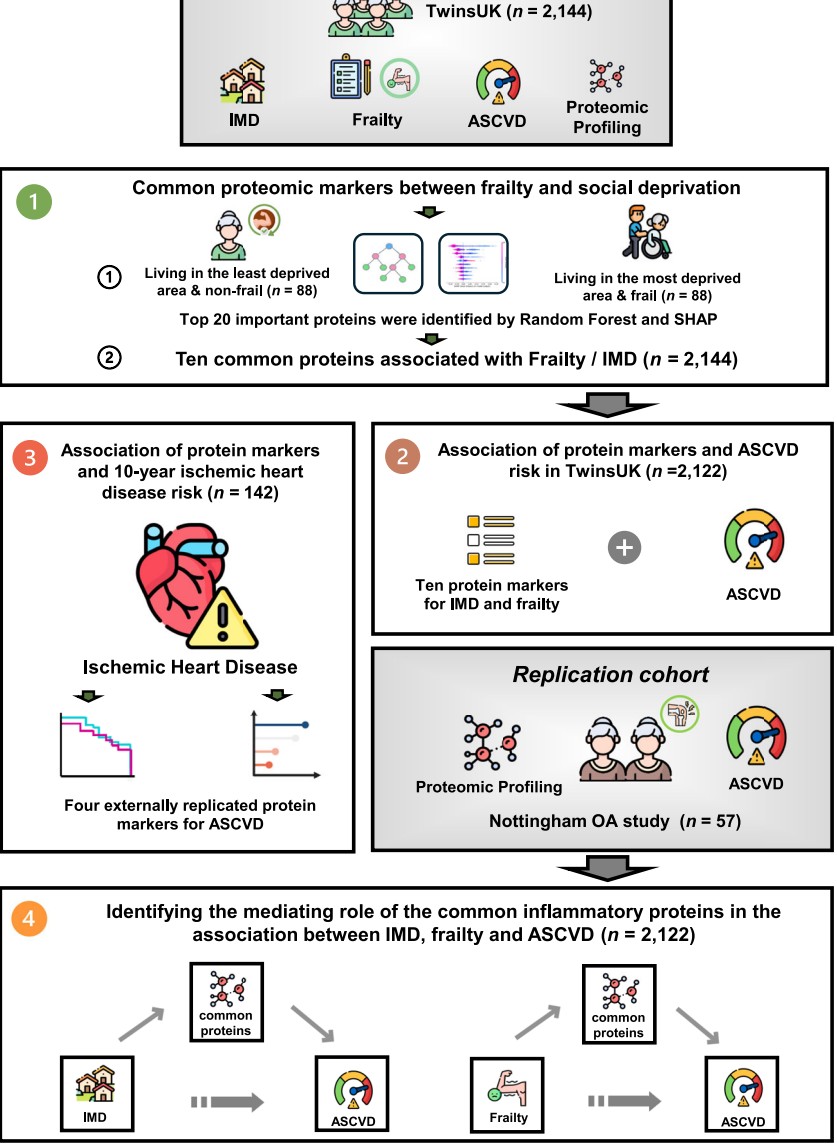

chemokines, growth factors, and enzymes[28,37]. The demographic characteristics of the study populations are reported in Table 1.

In TwinsUK, we identified 102 individuals living with frailty in disadvantaged areas and 151 without frailty and living in socioeconomically advantaged areas. The individuals who were both frail and living in the most deprived areas were older, had a higher BMI, lower level of education, a poorer diet, did less physical activity and smoked more than least deprived non-frail individuals (Supplementary Table 1).

Consistent with previous reports[38], we find that IMD is significantly associated with the FI after adjusting for age, BMI, and family relatedness (beta [95%CI] = −0.053 [−0.092, −0.014], P = 0.007). Results were consistent when further adjusting for age at cessation of full-time education, smoking, alcohol intake, dietary intake (healthy eating index, HEI), and physical activity (beta [95%CI] = −0.044 [−0.083, −0.005], p = 0.026).

**Common protein markers between frailty and social deprivation**
Random forest and SHAP analysis identified the 20 most important proteins from the inflammatory panel predictive of socially deprived frail and non-socially deprived non-frail conditions (Fig. 2A). Age-

matched individuals living in the most deprived areas with frailty (n = 88) and those in the least deprived areas without frailty (n = 88) were selected to construct the random forest model. A random forest model trained on the top 20 features achieved an AUC of 0.740 [interquartile range (IQR): 0.149], while a PLS-DA model outperformed the random forest model achieving an AUC of 0.797 [IQR: 0.043] (Supplementary Fig. 1).

To further interpret the most influential features of our RF model and understand their relative relationships, we tested their association with frailty and IMD using linear mixed models. The beta coefficients from the linear mixed model represent changes in the outcomes for each one-unit increase in NPX ($\log_2$ scale). After adjusting for age, BMI, batch effects, family relatedness and multiple testing, we found 16 proteins correlated with frailty and 11 proteins correlated with IMD, with an overlap of 10 proteins, including CDCP1, HGF, VEGFA, IL18R1, OSM, FGF21, TGFalpha, TNFSF14, CCL11, and FGF19 (FDR < 0.1) (Fig. 2B, Supplementary Data 1). Results were consistent when further adjusting for medication use (including those for blood pressure and lipid regulation and anti-diabetic treatments, Supplementary Fig. 2).

**Table 1 | Demographic characteristics of the study populations**

| Phenotype | TwinsUK | Nottingham OA |
|---|---|---|
| N | 2144 | 57 |
| Age, yrs | 60.14 (8.80) | 66.18 (7.77) |
| BMI, kg/m2 | 26.44 (4.69) | 29.92 (5.77) |
| IMD[a] | 7(4) | - |
| Frailty Index[a] | 0.11(0.12) | - |
| ASCVD risk score | 6.12 (7.01) | 15.18 (13.70) |
| Current smoker, N (%) | 190 (8.86%) | 0 (0%) |
| Individuals living with frailty in socioeconomically deprived areas, (%) | 102 (4.76%) | - |
| Non-frail individuals living in socioeconomically advantaged areas (5th quintile of IMD), N (%) | 151 (7.04%) | - |

The number in each cell denotes the mean (standard deviation, SD) for the continuous variables or count (percentages) for the categorical variables.
[a]Median (Interquartile Range, IQR).

### The link between inflammatory markers and cardiovascular risk

We then investigated the association between the ten inflammatory markers and the ASCVD risk score. After adjusting for age, BMI, batch effects and family relatedness, nine inflammatory markers including VEGFA, TNFSF14, TGFalpha, OSM, IL18R1, HGF, FGF21, CDCP1 and CCL11 were positively associated with ASCVD risk score (Fig. 3A and Supplementary Table 2, FDR < 0.1). Out of the nine protein markers associated with IMD, frailty, and ASCVD in TwinsUK, four were also associated with a higher risk of ASCVD in an independent cohort of 57 women with knee OA from the Nottingham OA study, including TNFSF14, HGF, CDCP1 and CCL11 (FDR < 0.1) (Fig. 3B and Supplementary Table 3). The association with IL18R1 had a consistent direction of effect although not statistically significant ($p = 0.052$).

To further understand the relationship between the validated inflammatory proteins and CVD risk, we investigated their association with 10-year risk of ischaemic heart disease. Age-matched controls were selected at baseline for individuals who developed incident ischaemic heart disease within 10 years. The Kaplan-Meier curve revealed a significant difference in ischaemic heart disease risk between groups with low versus high CDCP1 levels (Log-rank $p = 0.032$, Fig. 3C). The association between CDCP1 and ischaemic heart disease remained after further adjusting for BMI, batch effects, and family relatedness using mixed-effect Cox regression (Hazard Ratio [95% Confidence Interval]: 1.82 [1.17, 2.85], $p = 0.008$, Fig. 3D and Supplementary Table 4). Results were consistent when further adjusting for traditional risk factors including smoking, type 2 diabetes, use of blood pressure lowering medication, total and HDL cholesterol (HR = 1.68 [1.04–2.71], $P = 0.034$).

### Protein markers mediate the relationship between social deprivation and CVD risk

We then performed mediation analysis to test the role of the inflammatory proteins in the (i) IMD and ASCVD risk; and (ii) frailty and ASCVD risk. Among four markers, CDCP1, HGF and TNFSF14 mediated the association between IMD and ASCVD with VAF ranging from 11.4% to 27.0% (Fig. 4). Furthermore, these three inflammatory proteins also mediated the association between frailty and ASCVD with VAF ranging from 3.5% to 9.1% (Supplementary Fig. 3).

### Discussion

In this large population-based study we investigated 74 proteins from the Olink inflammation panel and identified ten proteins strongly associated with both frailty and area level social deprivation after adjusting for confounders[39]. Further analysis revealed that nine of these proteins were significantly associated with AHA ASCVD scores, pointing to a potential link between SES-related inflammation and CVD risk. These associations were validated in an independent cohort of middle-aged and older individuals with OA, with a set of four proteins being reproducibly associated. Importantly, an externally validated inflammatory marker CDCP1 was associated with higher risk of incident ischaemic heart disease. These results highlight the importance of addressing both inflammatory pathways and socioeconomic factors to reduce health disparities and improve outcomes in frail and deprived populations.

A considerable body of evidence shows that CVD development and progression are driven by chronic inflammation[1,2]. Indeed, several inflammatory cytokines have been reported in patients with heart disease and are associated with worse cardiovascular outcome[40,41]. We report that four of the ten proteins associated with both inflammaging and socioeconomic deprivation are also reproducibly linked to high CVD risk. This is consistent with deprivation playing a significant role in the incidence and outcome of heart disease[42]. Frailty, which is characterised by decreased physiological reserve and increased vulnerability to adverse health outcomes, has been previously shown to increase the risk of developing CVD[11,12]. Conversely, CVD can accelerate the progression of frailty due to the stress and damage inflicted on the cardiovascular system. Importantly, we report that TNFSF14, CCL11, CDCP1 and HGF are linked to both conditions and that HGF, CDCP1 and TNFSF14 might mediate the exacerbating effect of socioeconomic deprivation on these health conditions.

Additionally, the proteins identified reflect both specific socioeconomic–frailty pathways and general CVD inflammatory risk, underscoring the study's dual relevance to health disparities and cardiovascular medicine. This dual focus is a strength, advancing integration of social and biological perspectives in CVD research. The proteins linked to socioeconomic deprivation and frailty highlight actionable targets for interventions. If they mediate the impact of social determinants, they could guide public health interventions, personalised medicine approaches and mechanistic insights. For example, socioeconomic stress may exacerbate TNFSF14-driven adipose inflammation—a mechanism relevant to both specific and general contexts, which in turn fuels atherosclerosis[43,44]. This interdisciplinary approach provides a model for linking social determinants to biological mechanisms. The validation in an OA cohort suggests broader applicability, potentially to other comorbidities affected by inflammation.

Low-opportunity neighbourhoods often have limited access to healthcare services, exercise facilities, and high-quality educational resources, significantly impacting health behaviour and overall health, potentially leading to a decline in physical functioning. A longitudinal ageing cohort from Amsterdam reported persistently higher risk of frailty over 13 years in individuals with low education levels[45]. Moreover, HGF and TNFSF14 are negatively associated with both physical activity levels and healthy eating index, indicating that improving physical activity and dietary quality may effectively regulate these pro-inflammatory cytokines. Among the replicated protein markers associated with ASCVD risk, TNFSF14 (tumour necrosis factor ligand superfamily member 14) is produced by immune cells, substantially affecting adipose tissue phenotype[46]. A higher level of TNFSF14 has been reported in patients with Prader–Willi syndrome, a disease characterised by obesity and bone impairment[47]. In animal models, mice with a deficiency of TNFSF14 showed improved glucose tolerance and insulin sensitivity and decreased systemic inflammation and cytokine secretion in adipose tissue[48]. A prospective study investigated the clinical outcome of patients with coronary artery disease and found TNFSF14 was an independent predictor for the occurrence of mortality, myocardial infarction, and stroke[44]. Furthermore, a clinical trial on anti-inflammatory diets significantly reduced the level of TNFSF14 and systemic inflammation compared to Western diets in the control group[49]. The involvement of TNFSF14 in systemic inflammation and metabolic dysregulation reported in previous studies is consistent with TNFSF14 increasing CVD risk observed in our study. HGF (hepatocyte growth factor) was initially identified as a potent hepatocyte mitogen and an antiapoptotic factor for liver cells in the HGF/Met pathway which plays an important role in heart homoeostasis. The activation of the HGF/Met pathway in a transgenic mice model leads to

**Fig. 2 | The association of inflammation-related markers with area-level deprivation and frailty.** **A** The most important 20 proteins associated with socially deprived frail vs non socially deprived non frail conditions as identified using SHapley Additive exPlanation (SHAP) values from the Random Forest modelling. **B** The association of inflammatory markers with frailty index and IMD. Beta coefficients were calculated using linear mixed models adjusting for age and BMI with batch effects and family relatedness as random effects. Beta [95%CI] is reported for the association of frailty index and IMD with inflammatory proteins in the overall population (n = 2144).

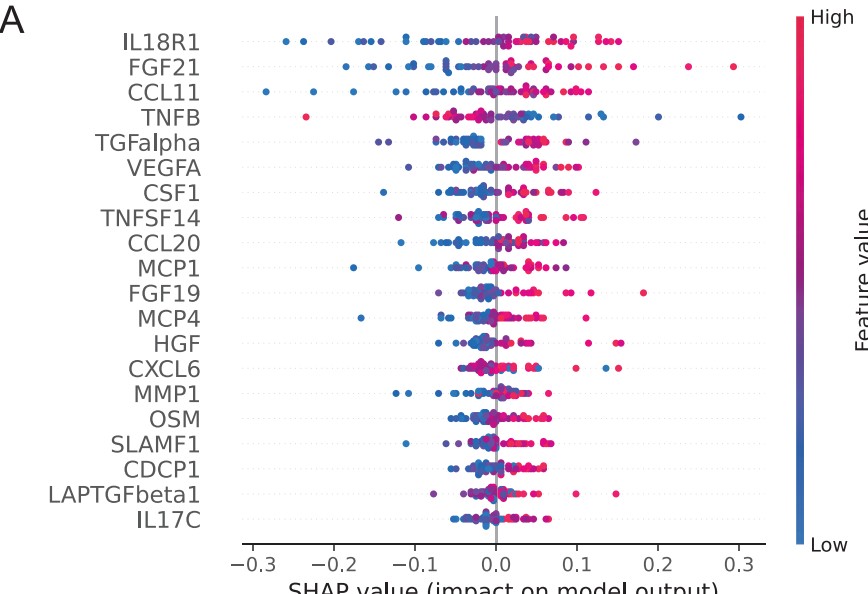

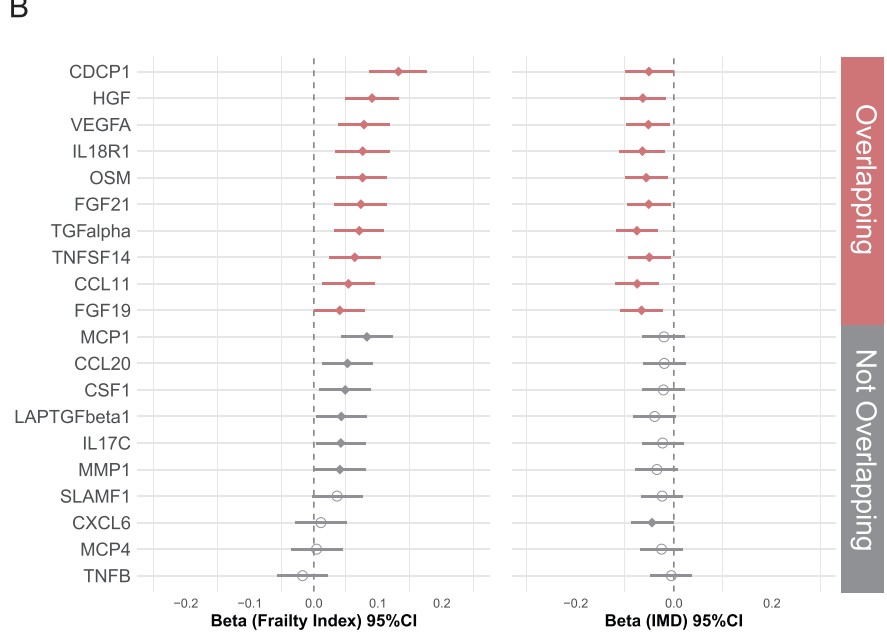

heart failure[50]. Similar to TNFSF14, diet has a large impact on HGF as the mice fed with a high-fat diet showed an increased level of HGF and obese mice switching to a control low-fat diet had a decreased level of HGF[51]. Additionally, higher HGF levels have been reported in patients with heart failure, progression of atherosclerosis, coronary heart disease and stroke[52]. CDCP1 (CUB domain containing protein 1) has been reported to be independently associated with higher risk of myocardial infarction[53] and is also positively associated with CHD and CVD mortality[54,55]. In addition, CDCP1 levels are elevated in individuals with type 1 diabetes[56] and soluble CDCP1 could serve as a potential disease marker in non-alcoholic steatohepatitis[57], demonstrating its close relationship with metabolic diseases. The association of these proteins with lower SES highlights how social determinants of health—such as poverty, lack of education, and limited access to healthcare—may increase both frailty and ASCVD risk via inflammatory pathways. Additionally, CCL11 (C-C motif chemokine 11) has been reported to promote the migration of vascular smooth muscle cells through CCR3 activation, indicating its role as a chemotactic factor in atherosclerotic plaque formation and arterial injury, potentially influencing

ASCVD progression[58]. A previous study showed that CCL11 was also involved in the recruitment of eosinophils which have a pathogenic effect in the development of myocarditis, while inhibiting this pathway has been demonstrated to have a protective effect against cardiac damage in mice models[59]. However, in our data we could not identify a mediatory role for CCL11 in the association between IMD and ASCVD, nor in the association between frailty and ASCVD.

This study benefits from the use of a large population-based cohort with in-depth phenotyping, longitudinal data, a comprehensive panel of inflammatory proteins and independent replication. The replication of these findings in individuals with OA suggests a broader applicability, advocating for integrated care approaches that consider both inflammatory status and social factors. Being a predefined panel, the Olink panel may not include all relevant inflammatory mediators for every context and the values delivered cannot compete in clinical usefulness with well-validated assays using established clinical thresholds. The strength of the panel lies in research and discovery, as shown in the study's identification of novel mediators like CDCP1. The discovery of associations between TNFSF14,

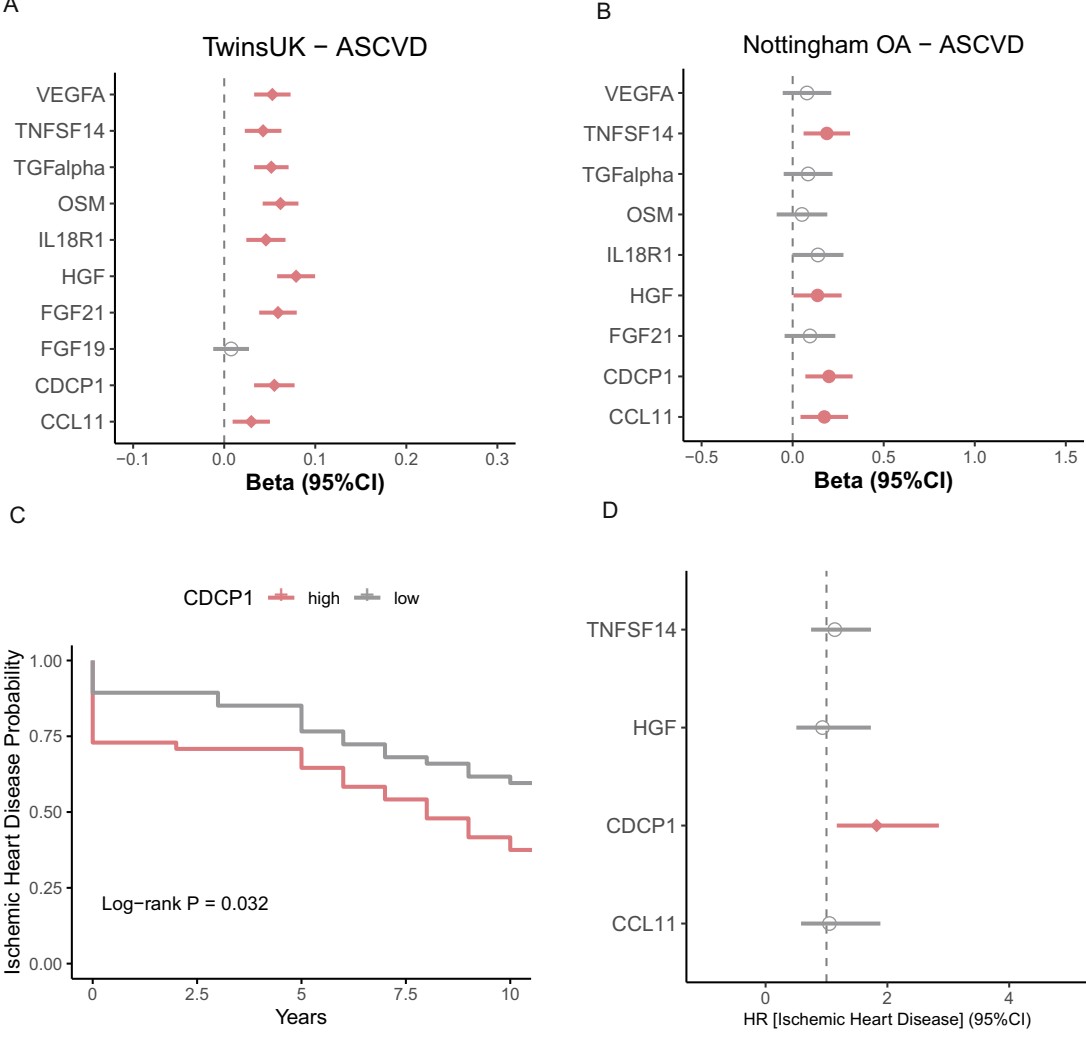

**Fig. 3 | The relationship between inflammatory markers and cardiovascular risk.** **A** The association of inflammation-related markers and ASCVD risk score. Beta coefficients were calculated using the linear mixed model adjusting for age, BMI and family relatedness. **B** External validation of association between inflammation-related markers and cardiovascular risk in the Nottingham OA cohort. Beta coefficients were calculated using the linear regression model, which adjusted for age and BMI. **C** Kaplan-Meier curve comparing the incidence of ischaemic heart disease risk

in TwinsUK participants with high versus low CDCP1 circulating levels (top and bottom tertiles). Age-matched controls were selected at baseline for individuals who developed ischaemic heart disease over a 10-year follow-up. **D** Association between externally validated inflammation-related markers and incidence of ischaemic heart disease risk in TwinsUK. Hazard Ratios (HRs) were calculated using mixed-effects Cox regression in age-matched cases and controls, adjusted for BMI, batch, and family relatedness.

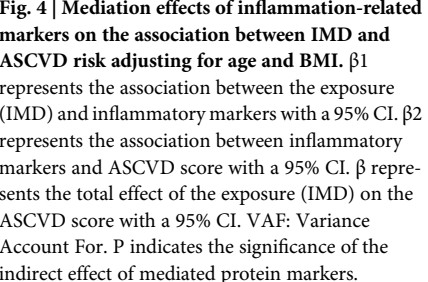

**Fig. 4 | Mediation effects of inflammation-related markers on the association between IMD and ASCVD risk adjusting for age and BMI.** β1 represents the association between the exposure (IMD) and inflammatory markers with a 95% CI. β2 represents the association between inflammatory markers and ASCVD score with a 95% CI. β represents the total effect of the exposure (IMD) on the ASCVD score with a 95% CI. VAF: Variance Account For. P indicates the significance of the indirect effect of mediated protein markers.

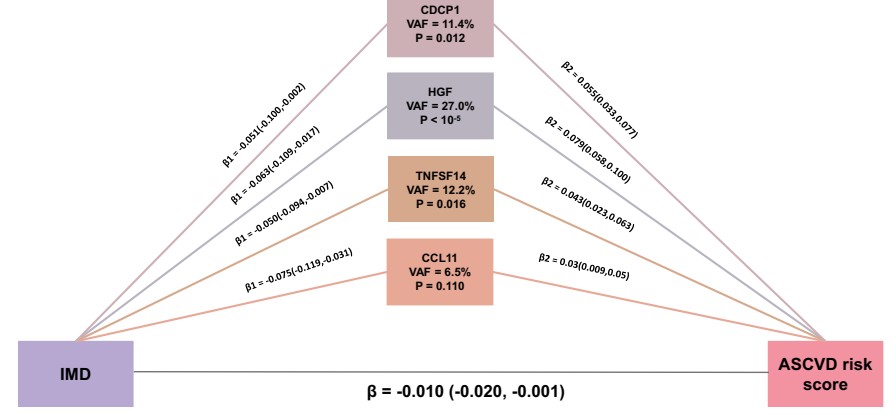

CCL11, CDCP1, HGF and frailty, ASCVD, and lower socioeconomic status underscores the complex interplay between inflammation, ageing, social determinants of health and CVD risk. Clinically, these proteins may serve as early biomarkers of risk, therapeutic targets, and guide personalised treatment. Although our results do not immediately alter daily clinical treatment, they can provide a foundation for future personalised medicine approaches. Clinicians could potentially develop clinical tests using these inflammatory proteins (particularly CDCP1, TNFSF14, and HGF) as biomarkers to identify patients at higher CVD risk, especially those who are frail or socioeconomically disadvantaged. This could guide earlier interventions, such as lifestyle modifications (e.g. improved diet, increased physical activity) or anti-inflammatory therapies. From a public health standpoint, the findings highlight the need for systemic interventions to reduce health disparities, focusing on improving conditions for vulnerable SES groups to lower the risk of both frailty and ASCVD.

We also note several limitations. Firstly, the results are primarily based on self-reported diet measurements and questionnaires, which may introduce reporting bias. Social deprivation was measured by neighbourhood deprivation level determined by postcode, which could misclassify the SES of participants. The Olink data only provided relative expression of proteins rather than absolute protein levels, restricting our ability to make definitive conclusions about the inflammatory responses. Other confounders, such as blood pressure, diabetes, and smoking habits, were not independently considered; however, the ASCVD score used in this study incorporates systolic blood pressure, diabetes diagnosis, the use of hypertension medication, and smoking status, which may partially address these factors. We replicated our findings in an OA cohort, and the effect size estimates may not be reflective of those in the general population. However, OA affects 40–60% of individuals after age 60[60,61]. the prevalence of OA is significantly higher (as much as 50%) among socioeconomically deprived individuals than among those who do not experience deprivation[62,63]. Therefore, the replication cohort is representative of a substantial proportion of the elderly population. We have thus tested the role of inflammatory markers linked to deprivation in a group of people more likely to develop CVD, to develop frailty and to suffer deprivation. Our findings were discovered and replicated predominantly in female participants of European ancestry across two independent cohorts, which may limit the generalisability of the results to broader populations. Future studies should include a more diverse population with different ethnic backgrounds and both male and female individuals, ensuring sufficient sample sizes to validate these findings comprehensively. Furthermore, the study's findings of inflammatory markers require further validation before routine clinical adoption, and no specific treatments targeting these proteins are currently recommended based solely on this research.

In conclusion, our results underscore the importance of addressing social determinants of health to reduce inflammation-related frailty, emphasising the need for policies that mitigate social deprivation. The link between inflammatory markers and CVD risk highlights potential for early intervention in at-risk populations to prevent or delay the onset of both frailty and CVD, leading to improved health outcomes and reduced healthcare costs. These findings confirm the physiologically important role of inflammatory pathways in frailty status and CVD risk, offering a promising target for reducing disparities in healthcare in socially disadvantaged and vulnerable groups.

## Data availability
The data used in this study are held by the Department of Twin Research at King's College London. The data can be released to bona fide researchers using our normal procedures overseen by the Wellcome Trust and its guidelines as part of our core funding (http://twinsuk.ac.uk/resources-for-researchers/access-our-data/). Data from the Nottingham OA cohort is available upon reasonable request from the principal investigator (ana.valdes@nottingham.ac.uk). Supplementary Data 1 is the source data for Fig. 2B. Supplementary Table 2 is the source data for Fig. 3A. Supplementary Table 3 is the source data for Fig. 3B. Supplementary Table 4 is the source data for Fig. 3D.

## Code availability
The key source codes that supported these results and findings can be found in zenodo.org (https://zenodo.org/records/15670583)[64].

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

## Acknowledgements

This research was funded in whole, or in part, by the UKRI (MR/Y010175/1, MR/T004142/1) and by the Wellcome Leap Dynamic Resilience Programme (co-funded by Temasek Trust). For the purpose of open access, the authors have applied a CC BY public copyright to any Author Accepted Manuscript version arising from this submission. The Department of Twin Research received support from grants from the Wellcome Trust (212904/Z/18/Z) and the Medical Research Council (MRC)/British Heart Foundation (BHF) Ancestry and Biological Informative Markers for Stratification of Hypertension (AIM-HY; MR/M016560/1), European Union, Chronic Disease Research Foundation (CDRF), Zoe Global Ltd., the NIHR Clinical Research Facility and Biomedical Research Centre (based at Guy's and St Thomas' NHS Foundation Trust in partnership with King's College London). The TwinsUK study was also supported by grant funding from Tinnitus UK. C.M. is funded by the Chronic Disease Research Foundation (CDRF), by the Italian Ministry of Education and Research (MUR): Dipartimenti di Eccellenza Program 2023-2027 and by the Italian Ministry of Health – Bando Ricerca Corrente. AMV is supported by the National Institute for Health and Care Research Nottingham Biomedical Research Centre. Support for this work was also provided by UKRI/MRC grants (MR/W026813/1 and MR/Y010175/1) awarded to A.M.V. and C.M. M.N.L. is supported by a National Institute of Health Research fellowship (NIHR303488).

## Author contributions

Conceived and designed the experiments: C.M., A.M.V. Contributed reagents/materials/analysis tools: R.C.E.B., M.N.L., F.M.K.W., C.J.S. Data Curation: N.R., M.F. Formal analysis: Y.L., P.L., A.M.V., C.M. External replication: A.K., A.K., V.G. Wrote the manuscript: Y.L., P.L., A.M.V., C.M. Revised the manuscript: all authors. All authors reviewed and approved the final version of the manuscript.

## Competing interests

A.M.V. is a consultant of ZOE Global Ltd. All other authors declare no competing financial interests.
