## [Transparent Peer Review file · Communications Medicine]

Common inflammatory proteins linking frailty and area-level deprivation as key drivers of cardiovascular risk

Corresponding Author: Dr Cristina Menni

Version 0:

Reviewer comments:

Reviewer #1

(Remarks to the Author)

This is an interesting protein analysis using the TwinsUK participants.

I will focus on the statistics, machine learning and proteomics and will leave the relevance of the research question to other reviewers. Though I found the paper interesting and clear to read. The analysis is well done and the authors are sufficiently careful for example using group splitting for their machine learning models to avoid leakage and an overestimating performance. I have some minor comments that would improve the clarity in places.

1. I would not really call the olink a proteomics platform. I think using the proteomics is a bit confusing as I wondered why the number of proteins is so small. I think "Olink panel" is sufficient to describe the experiment that was performed.
2. It was difficult to understand early on the manuscript what olink platform was actually measuring - perhaps it worth describing this early in the results (briefly) to contextualise the results for the reader. I assume all the effect sizes correspond to changes in abundance or log2 abundance, but I was unsure.
3. It would be nice to have a table of all the statistical results with coefficients and associated p-values.
4. It's unclear how the confidence intervals were calculated - normal assumption, bootstrapping? It would be good to spell this out in the methods.
5. It was also unclear how the mediation analysis and survival analysis was performed. I assume these were standard applications of the methods but it would be good to know this was done correctly by describing the methods.

Reviewer #2

(Remarks to the Author)

In this study by Lin et al., data from the TwinsUK registry were analyzed to determine the inter-relationships between frailty, social deprivation, inflammation, and cardiovascular disease risk. The results identified nine biologically plausible inflammation proteins that were associated with frailty, social deprivation, and 10-year ASCVD risk scores. Among these proteins, four (TNFSF14, HGF, CDCP1, and CCL11) were similarly associated with a higher ASCVD risk score in female participants of the Nottingham OA study. Further analyses in TwinsUK identified that CDCP1, HGF, and TNFSF14 mediated the associations of both social deprivation and, separately, frailty with higher ASCVD risk score. Finally, CDCP1 was associated with an increased risk of incident ischemic heart disease over 10-years of study follow-up. The study is carefully designed and clearly presented. The results provide some biological insights into potential mechanism(s) underlying the relationships between frailty, socioeconomic status, and CVD risk. Comments and questions for the authors consideration are below.

1.) Figure 2a presents results from a SHAP analysis with the 20 most important proteins associated with socially deprived and frail as compared with non-socially deprived and non-frail categories. How were the number of "important" proteins

(n=20) chosen?

2.) In the results presented in Figure 2b, the 20 proteins that were determined in Figure 2a to be differentially associated with area-level deprivation and frailty were analyzed in separate linear regression models with IMD and frailty as separate outcomes. Can the authors provide text in the methods or results clarifying how these two sets of analyses (those in Figs 2a and 2b) inform one another or provide complimentary information?

3.) The results in Figure 2b are stated to be adjusted for batch, whereas the results in figure 2a are not specified to be adjusted for batch. Please clarify.

4.) Throughout the analyses, statistical significance is defined as an FDR p-value <0.10. Was this p-value specified a priori? The authors should provide a rationale for this threshold.

5.) The inclusion of a replication cohort adds rigor to the study design. However, the replication cohort included a smaller sample of women with knee osteoarthritis (OA), which was not matched for in TwinsUK. This raises the question as to whether there was an influence of OA on the magnitude of the association among the nine proteins tested in the replication analyses and it is not possible to evaluate the influence of OA in the analyses.

6.) Although acknowledged as a limitation in the Discussion, the analysis of CDCP1 with incident ischemic heart disease is not adjusted for traditional CVD risk factors which limits the inference of this result. Are CVD risk factors available in TwinsUK that could be included as model covariates?

7.) How was incident ischemic heart disease adjudicated?

8.) The Discussion cites prior literature reporting associations of TNFSF14, HGF, and CDCP1 with increased risk of CVD events. This literature provides overall support for the current findings. However, as these proteins were not specific to frailty or social deprivation in these prior studies and were found to mediate the association between both frailty and social deprivation, separately, with ASCVD risk score in the current study, it is difficult to determine whether these proteins represent specific biological pathways underlying SDOH-related frailty or are more broadly identifying the contribution of inflammation to CVD risk as may be related to a multitude of CVD risk factors.

Minor:

Presenting median (IQR) values in Table 1 would be informative to help understand the heterogeneity of participants' frailty and IMD status.

Typographical error: Page 17, line 353: "We "run" a mediation analysis".

Reviewer #3

(Remarks to the Author)
TO THE EDITORS

The study entitled
Common inflammatory proteins linking frailty and area-level deprivation as key drivers of cardiovascular risk is interesting and provides new informations in this topic
However, My concerns are regarding the study population, methodology, other risk factors which may affect the results
I suggest for these reasons to be reject

Version 1:

Reviewer comments:

Reviewer #1

(Remarks to the Author)
The authors have responded to my concerns.

Reviewer #2

(Remarks to the Author)
The authors have thoughtfully and carefully addressed my prior questions and suggestions. No additional comments.

Reviewer #3

(Remarks to the Author)

This is an interesting study assessing Common inflammatory proteins linking frailty and area-level deprivation as key drivers of cardiovascular risk. After the extensive revision My concerns remains as the following

- 1.The clarity of the study is uncertain
2. The clinical utility remains unclear
3. The daily practice based on study findings remains unanswerable

Line numbers in this letter refer to the line numbers within the clean version of the manuscript (IMD_frailty_proteins_clean.docx).

Editorial comments:

1. Please address the concerns raised by Referee #1, and the limitations of the replication cohort as noted by Referee #2, clearly explaining how this affects interpretation. We strongly encourage you to validate your findings in a new matched replication cohort.

Author's response: We agree that having an additional cohort would make results even more robust, but unfortunately we do not have access to other cohorts with the clinical and biochemical measures needed to assess ASCVD, in addition to SES (IMD) and Olink panel profiling. We have addressed this point in reply to point #8 by referee #2 and we believe that the use of an OA cohort can be viewed as a strength given the study objectives. We have clearly stressed the lack of generalisability and the caveats of the study design in greater detail now in the Study Limitations section.

2. Additionally, we expect a clear discussion on whether the identified proteins reflect specific socioeconomic factors—frailty pathways or more broadly indicate inflammatory risk for CVD (Referee #2), and how this distinction affects the overall advance of your work.

Authors' response: Thank you for this comment. We have now added the following text to the Discussion in lines 187-197:

“Additionally, the proteins identified reflect both specific socioeconomic–frailty pathways and general CVD inflammatory risk, underscoring the study’s dual relevance to health disparities and cardiovascular medicine. This dual focus is a strength, advancing integration of social and biological perspectives in CVD research. The proteins linked to socioeconomic deprivation and frailty highlight actionable targets for interventions. If they mediate the impact of social determinants, they could guide public health interventions, personalised medicine approaches and mechanistic insights.

For example, socioeconomic stress may exacerbate TNFSF14-driven adipose inflammation—a mechanism relevant to both specific and general contexts, which in turn fuels atherosclerosis^{27,28}.

This interdisciplinary approach provides a model for linking social determinants to biological mechanisms. The validation in an osteoarthritis cohort suggests broader applicability, potentially to other comorbidities affected by inflammation.”

Reviewer #1 (Remarks to the Author):

This is an interesting protein analysis using the TwinsUK participants.

I will focus on the statistics, machine learning and proteomics and will leave the relevance of the research question to other reviewers. Though I found the paper interesting and clear to read. The analysis is well done and the authors are sufficiently careful for example using group splitting for their machine learning models to avoid leakage and an overestimating performance. I have some minor comments that would improve the clarity in places.

Authors' response: We thank the reviewer for the positive evaluation of our manuscript.

1. I would not really call the olink a proteomics platform. I think using the proteomics is a bit confusing as I wondered why the number of proteins is so small. I think "Olink panel" is sufficient to describe the experiment that was performed.

Authors' response: We thank the reviewer for the suggestion. We have changed the term "proteomics" to "Olink panel", and "proteomic marker" to "inflammation-related marker" or "inflammatory marker" throughout the main text and supplementary materials to enhance clarity and accuracy.

2. It was difficult to understand early on the manuscript what olink platform was actually measuring - perhaps it worth describing this early in the results (briefly) to contextualise the results for the reader. I assume all the effect sizes correspond to changes in abundance or log2 abundance, but I was unsure.

Authors' response: We apologise to the reviewer for the lack of clarity. We have now described what the Olink platform actually measures at the beginning of the results' section.

Lines 61-65:

"We included 2,144 females from TwinsUK¹⁷, and 57 middle-aged to older female individuals with knee OA from the Nottingham OA study^{18,19}, with protein data from the Olink Target 96 Inflammation panel. This panel uses proximity extension assay technology to sensitively quantify inflammation-related proteins, including cytokines, chemokines, growth factors, and enzymes^{20,21}"

As the proteins are measured using normalised protein expression values on a log₂ scale, we have clarified that the effect sizes of the linear mixed model correspond to changes in log₂ abundance.

Lines 95-97

“The beta coefficients from the linear mixed model represent changes in the outcomes for each one-unit increase in Normalised Protein Expression (log₂ scale)”

3. It would be nice to have a table of all the statistical results with coefficients and associated p-values.

Authors’ response: We have now included tables with all statistical results with coefficients and associated p-values in the Supplementary Tables 2-5.

4. It's unclear how the confidence intervals were calculated - normal assumption, bootstrapping? It would be good to spell this out in the methods.

Authors’ response: The 95% confidence intervals for effect sizes were calculated using the R function `confint()`, which provides more estimates using profile likelihood-based confidence intervals for fixed effects. We added the description for the confidence interval calculation to the method in lines 364-366.

5. It was also unclear how the mediation analysis and survival analysis was performed. I assume these were standard applications of the methods but it would be good to know this was done correctly by describing the methods.

Authors’ response: We apologise to the reviewer for the lack of clarity. We have expanded our Methods section to better describe both analyses. We confirmed all standard assumptions for the mediation analysis. We used bootstrapping with 1000 simulations to derive confidence intervals for the indirect effects.

The details were added to the Supplementary Text Section:

“We conducted causal mediation analyses following the Baron and Kenny framework¹. First, we evaluated the three essential mediation assumptions: (1) significant association between the independent variable and the dependent variable, (2) significant association between the independent variable and the mediator, and (3) significant association between the mediator and the

dependent variable when controlling for the independent variable. After confirming these assumptions, we implemented formal causal mediation analysis using the 'mediate' function from the R package 'mediation' (version 4.5.0)². Each mediator was analysed independently. We determined significant mediation based on both statistical significance ($p < 0.05$) and the magnitude of the indirect effect. To quantify the strength of mediation, we calculated the variance accounted for (VAF) as the ratio of indirect-to-total effect. This metric represents the proportion of variance explained by the mediation pathway - specifically, the proportion of the effect of IMD on frailty transmitted through inflammation-related proteins.”

We further clarify the details for survival analysis in lines 403-415 in the Methods Section.

“For the survival analysis, the time-to-event outcome was incident ischemic heart disease. Cases were defined as participants diagnosed with ischemic heart disease within 10 years following the collection of Olink inflammatory data. Individuals with a prior history of ischemic heart disease or who developed the disease during follow-up were excluded from the control group. Age-matched controls were selected at baseline for subjects with ischemic heart disease. The Kaplan-Meier method was used to estimate the probability of ischemic heart disease for individuals in the top and bottom tertiles of four replicated protein markers. Log-rank tests were performed to assess whether there was a statistically significant difference in disease risk between groups. To evaluate the impact of other covariates, including BMI and other CVD-related factors (smoking status, diabetes diagnosis, use of anti-hypertensive medication, total cholesterol level, HDL cholesterol level), on ischemic heart disease risk, we used mixed-effects Cox regression models to estimate the association between externally validated markers and 10-year ischemic heart disease risk, accounting for batch effects, family relatedness, and potential confounders.”

Reviewer #2 (Remarks to the Author):

In this study by Lin et al., data from the TwinsUK registry were analyzed to determine the inter-relationships between frailty, social deprivation, inflammation, and cardiovascular disease risk. The results identified nine biologically plausible inflammation proteins that were associated with frailty, social deprivation, and 10-year ASCVD risk scores. Among these proteins, four (TNFSF14, HGF, CDCP1, and CCL11) were similarly associated with a higher ASCVD risk score in female participants of the Nottingham OA study. Further analyses in TwinsUK identified that

CDCP1, HGF, and TNFSF14 mediated the associations of both social deprivation and, separately, frailty with higher ASCVD risk score. Finally, CDCP1 was associated with an increased risk of incident ischemic heart disease over 10-years of study follow-up. The study is carefully designed and clearly presented. The results provide some biological insights into potential mechanism(s) underlying the relationships between frailty, socioeconomic status, and CVD risk.

Authors' response: We thank the reviewer for the positive evaluation of our manuscript.

Comments and questions for the authors consideration are below.

1. Figure 2a presents results from a SHAP analysis with the 20 most important proteins associated with socially deprived and frail as compared with non-socially deprived and non-frail categories. How were the number of "important" proteins (n=20) chosen?

Authors' response: We thank the reviewer for this valuable comment. The selection of the top 20 proteins from the SHAP analysis follows a widely used rule of thumb, aligning with the default for the important feature display in SHAP-based visualisations. The use of a top-20 cut-off has also been commonly adopted in other omics studies employing SHAP for feature ranking. For example, several published studies¹⁻³ have utilised a similar approach, selecting the top 20 features to balance interpretability and biological relevance while avoiding overfitting. We have now included this clarification in the revised manuscript (see [Methods Section, lines 382-384]).

2. In the results presented in Figure 2b, the 20 proteins that were determined in Figure 2a to be differentially associated with area-level deprivation and frailty were analyzed in separate linear regression models with IMD and frailty as separate outcomes. Can the authors provide text in the methods or results clarifying how these two sets of analyses (those in Figs 2a and 2b) inform one another or provide complimentary information?

Authors' response: We apologise for the lack of clarity. The results of Figure 2a and 2b are complementary. Indeed, we used random forest to identify candidate proteins, which were then validated and refined through covariate-adjusted modelling.

We have now added the sentence below to the Methods Section in lines 367-372 to clarify

“We used a two-step approach to identify proteins associated with deprivation and frailty. We first employed a Random Forest model with Shapley Additive exPlanations (SHAP)⁶⁰ to identify the 20 most predictive proteins (based on feature importance) for distinguishing frail individuals in deprived areas from non-frail individuals in advantaged areas. We then tested these proteins individually using linear mixed models, with deprivation and frailty as separate outcomes, adjusting for age, BMI, batch, and family structure.”

3. The results in Figure 2b are stated to be adjusted for batch, whereas the results in figure 2a are not specified to be adjusted for batch. Please clarify.

Authors’ response: We thank the reviewer for this comment. We have now clarified that we did not include any covariates in the analysis displayed in Figure 2A as we just wanted to identify the most important proteins in Random Forest analysis. In Figure 2B, results are adjusted for covariates including batch as we wanted to see whether the proteins identified as part of the RF algorithm remained significant after controlling for confounders.

4. Throughout the analyses, statistical significance is defined as an FDR p -value < 0.10 . Was this p -value specified a priori? The authors should provide a rationale for this threshold.

Authors’ response: Using FDR < 0.10 aligns with the exploratory, high-dimensional nature of omics research, and is supported by biological plausibility, external validation, and study design safeguards. This threshold enables the identification of novel inflammatory mediators (e.g., CDCP1) while maintaining scientific rigor through subsequent validation, making it a justified choice for advancing the study’s aims of uncovering links between socioeconomic deprivation, frailty, and CVD risk. The use of FDR < 0.1 is also commonly accepted in other omics studies^{4,5}.

5. The inclusion of a replication cohort adds rigor to the study design. However, the replication cohort included a smaller sample of women with knee osteoarthritis (OA), which was not matched for in TwinsUK. This raises the question as to whether there was an influence of OA on the magnitude of the association among the nine proteins tested in the replication analyses and it is not possible to evaluate the influence of OA in the analyses.

Authors’ response: We thank the reviewer for raising this issue. As discussed in reply to point 8 below, OA is more prevalent among socioeconomically deprived groups, and both CVD and frailty are more prevalent in individuals with OA than in those without. We agree, and have made this clear in the Study limitations, that this might result in effect size estimates that are not reflective of those in the general population. However, the aim of this study was to identify

mediators of socioeconomic deprivation on the intersection between CVD and frailty not to develop a diagnostic with reproducible effect sizes. The OA cohort thus is suitable for the scope for which we've used it. We agree that further research in other populations is needed to take this research forward.

6. Although acknowledged as a limitation in the Discussion, the analysis of CDCP1 with incident ischemic heart disease is not adjusted for traditional CVD risk factors which limits the inference of this result. Are CVD risk factors available in TwinsUK that could be included as model covariates?

Authors' response: We appreciate the reviewer's comment regarding the need to adjust for traditional CVD risk factors in the analysis of CDCP1 and incident ischemic heart disease. As the ASCVD score was included in our analysis and its components are widely recognised as traditional CVD risk factors, all of which are available in the TwinsUK cohort, we incorporated these variables as covariates in fully adjusted mixed-effects Cox regression models. The covariates for the model include smoking status, diabetes diagnosis, use of anti-hypertensive medication, total cholesterol level, HDL cholesterol level and also BMI, batch and family relatedness. In the fully adjusted model, the association between CDCP1 and incident ischemic heart disease was still statistically significant (HR = 1.68 [1.04–2.71], p = 0.034) in the age-matched cases and controls.

The results for the fully adjusted model were added to the main text in lines 133-135.

7. How was incident ischemic heart disease adjudicated?

Authors' response: We have now clarified that incident ischemic heart disease was ascertained through self-reported questionnaires administered to participants at regular follow-up intervals. Specifically, participants were asked whether a doctor had formally diagnosed them with IHD since their previous assessment. We defined incident cases as those reporting a diagnosis of IHD during the follow-up period who had not reported such a diagnosis at baseline. While we acknowledge the use of self-reported outcomes in our limitations, this approach is consistent with methods used in other large-scale epidemiological studies. We have added the ascertainment of incident ischemic heart disease in lines 332-336.

8. The Discussion cites prior literature reporting associations of TNFSF14, HGF, and CDCP1 with increased risk of CVD events. This literature provides overall support for the current findings. However, as these proteins were not specific to frailty or social deprivation in these prior studies and were found to mediate the association between both frailty and social deprivation, separately, with ASCVD risk score in the current study, it is difficult to determine whether these proteins represent specific biological pathways underlying SDOH-related frailty or are more broadly identifying the contribution of inflammation to CVD risk as may be related to a multitude of CVD risk factors.

Authors' response: We agree that we replicated our findings in an OA cohort, and the effect size estimates may not be reflective of those in the general population. However, OA affects 40%-60% of individuals after age 60^{6,7} and the prevalence of OA is significantly higher (as much as 50%) among socioeconomically deprived individuals than among those who do not experience deprivation^{8,9}. Therefore, the replication cohort is representative of a very large proportion of the elderly population. Testing the role of inflammatory markers linked to deprivation in a group of people more likely to develop CVD, to develop frailty and to suffer deprivation is highly relevant. We have addressed the lack of generalisability more clearly now in the *Study Limitations* section.

Minor:

Presenting median (IQR) values in Table 1 would be informative to help understand the heterogeneity of participants' frailty and IMD status.

Authors' response: We thank the reviewer for the suggestion. We have now added median (IQR) to Table 1.

Typographical error: Page 17, line 353: "We "run" a mediation analysis".

Authors' response: Thanks for spotting the typo. We have now corrected it in line 418.

Reviewer #3 (Remarks to the Author):

The study entitled
Common inflammatory proteins linking frailty and area-level deprivation as key drivers
of cardiovascular risk is interesting and provides new informations in this topic

We thank the reviewer for the positive evaluation of our manuscript.

My concerns are the following

1. What is the clinical message of the study?

Authors' response: We thank the reviewer for this comment. We have now clarified in the discussion that these results highlight the importance of addressing both inflammatory pathways and socioeconomic factors to reduce health disparities and improve outcomes in frail and deprived populations. We added this to the Discussion Section in lines 172-174.

2. How the results affect the daily clinical treatment of the patients?

Authors' response: Although our results do not immediately alter daily clinical treatment, they can provide a foundation for future personalised medicine approaches. Clinicians could use these inflammatory proteins (particularly CDCP1, TNFSF14, and HGF) as biomarkers to identify patients at higher CVD risk, especially those who are frail or socioeconomically disadvantaged. This could guide earlier interventions, such as lifestyle modifications (e.g., improved diet, increased physical activity) or anti-inflammatory therapies. However, the study's findings of inflammatory markers require further validation before routine clinical adoption, and no specific treatments targeting these proteins are currently recommended based solely on this research.

We have incorporated the above content in the Discussion Section in lines 250-256 and lines 278-281.

3. How you select these inflammatory mediators?

Authors' response: We have now clarified that we used a two-step approach the Methods Section in lines 367-372. We used a two-step approach to identify proteins associated with deprivation and frailty. We first employed a Random

Forest model with SHapley Additive exPlanations (SHAP)¹⁰ to identify the 20 most predictive proteins (based on feature importance) for distinguishing frail individuals in deprived areas from non-frail individuals in advantaged areas. We then tested these proteins individually using linear mixed models, with deprivation and frailty as separate outcomes, adjusting for age, BMI, batch, and family structure.

4. You need to provide 1-2 of the most important mediators

Authors' response: We have now clarified that the most important mediators identified are TNFSF14, HGF, and CDCP1, as they significantly mediated the associations between IMD and ASCVD risk (variance accounted for: 11.4%–27.0%) and between frailty and ASCVD risk (variance accounted for: 3.5%–9.1%). CDCP1 was particularly notable, showing a significant association with incident ischemic heart disease risk (HR 1.82 [95% CI 1.17, 2.85]). We have highlighted this in line 184-186.

5. The relative expression of proteins does not provide a clear picture of the inflammatory process

Authors' response: We thank the reviewer for this comment. The Olink panel captures a wide range of cytokines, chemokines, and other inflammatory mediators (e.g., TNFSF14, HGF, CDCP1, CCL11), allowing for a broader assessment of inflammatory pathways involved in complex conditions like frailty, socioeconomic deprivation, and cardiovascular disease (CVD) risk. This multi-protein approach can reveal nuanced associations and potential mediators that a narrow focus on single markers of systemic inflammation commonly used in clinical practice (e.g. hsCRP) might miss, as such markers primarily reflect acute-phase responses and general inflammation rather than the full spectrum of inflammatory processes.

We acknowledge that, while the Olink panel is more informative, as being a predefined panel, the Olink panel may not include all relevant inflammatory mediators for every context, and the values delivered cannot compete in clinical usefulness with well-validated assays using established clinical thresholds. The strength of the panel lies in research and discovery, as shown in the study's identification of novel mediators like CDCP1, but it's not yet a replacement for targeted markers in daily practice, as we now acknowledge in the Discussion of our findings in lines 243-246.

6. What about the effects of other risk factors and the medications-These may affect the results

Authors' response: Thank you for the helpful comment. We have considered the potential confounding effects of medications prescribed for diabetes, hypertension, and hyperlipidaemia, as these are relevant to both CVD risk and frailty. Medication usage data for all participants are presented in Supplementary Table 1, showing higher use among the frail and most deprived group. These medications were included as covariates in the linear mixed models assessing protein associations with frailty and deprivation. While effect sizes were slightly attenuated, all previously significant proteins remained so (Supplementary Figure 2). For the ASCVD analysis, we did not adjust for medication use, as relevant components—such as hypertension treatment, diabetes status, and lipid levels—are already incorporated into the ASCVD risk score, and further adjustment would risk multicollinearity. This clarification and the updated results are now included in the supplementary file and in the Results Section in lines 101-103.

Supplementary Figure 2. The association of inflammation-related markers with frailty index and IMD. Beta coefficients were calculated using linear mixed models adjusting for age, BMI, medications for hypertension, diabetes, and hyperlipidaemia, with family relatedness and batch as random effects. Beta [95%CI] is reported for the association of frailty index and IMD with protein signatures in the overall population (n = 2,144). The black font in the y-axis represents the overlapping protein markers for frailty and IMD, and the grey font represents the non-overlapping protein markers.

7. Is the study population the ideal FOR THIS TYPE OF STUDY. What about to select a general population and a spread of age and to include male and female

Authors' response: We thank the reviewer for this comment. TwinsUK individuals have a wide age range (in this study, the age range is 37 to 84 yrs). We do agree that the lack of male participants is a limitation. However, TwinsUK is for historic reasons female predominant. Future studies should confirm these findings in males. We have highlighted this important limitation in the discussion.

Rebuttal letter references (citations within quotes reflect the reference list of the manuscript)

1. Zhuang, Y. *et al.* Deep learning on graphs for multi-omics classification of COPD. *PLOS ONE* **18**, e0284563 (2023).
2. Wang, P. *et al.* Prediction of plant complex traits via integration of multi-omics data. *Nature Communications* **15**, 1–15 (2024).
3. Bai, Z. *et al.* Predicting response to neoadjuvant chemotherapy in muscle-invasive bladder cancer via interpretable multimodal deep learning. *NPJ Digit Med* **8**, 174 (2025).
4. Priya, S. *et al.* Identification of shared and disease-specific host gene-microbiome associations across human diseases using multi-omic integration. *Nat Microbiol* **7**, 780–795 (2022).
5. Cheng, Y. *et al.* Perturb-tracing enables high-content screening of multi-scale 3D genome regulators. *Nat Methods* (2025) doi:10.1038/s41592-025-02652-z.
6. Zhang, Y. & Jordan, J. M. Epidemiology of osteoarthritis. *Clin Geriatr Med* **26**, 355–369 (2010).
7. Ho-Pham, L. T. *et al.* Prevalence of radiographic osteoarthritis of the knee and its relationship to self-reported pain. *PLoS One* **9**, e94563 (2014).

8. Long, H. *et al.* Prevalence Trends of Site-Specific Osteoarthritis From 1990 to 2019: Findings From the Global Burden of Disease Study 2019. *Arthritis Rheumatol* **74**, 1172–1183 (2022).
9. Elgaddal, N., Kramarow, A. E., Weeks, D. J. & Reuben, C. Arthritis in Adults Age 18 and Older: United States, 2022.
<https://www.cdc.gov/nchs/products/databriefs/db497.htm> (2024).
10. Lundberg, S. & Lee, S.-I. A Unified Approach to Interpreting Model Predictions. (2017).

Editorial comment

It is our editorial view that the comments from Reviewer #3 have been addressed in the revised manuscript. However, we suggest you highlight the clinical message (including the clinical need/utility/daily practice) in the Introduction section as well.

Authors' response: We thank the editor for this comment. We have now highlighted the clinical message (including the clinical need/utility/daily practice) in the Introduction section as well.

Identifying these proteins as early biomarkers could enable risk stratification, guide personalised treatments, and highlight therapeutic targets. Clinically, such biomarkers may inform future tests to identify frail or socioeconomically disadvantaged patients at elevated CVD risk due to exacerbated inflammation and aging.

Reviewer #3 (Remarks to the Author):

This is an interesting study assessing Common inflammatory proteins linking frailty and area-level deprivation as key drivers of cardiovascular risk. After the extensive revision My concerns remains as the following

1.The clarity of the study is uncertain

Authors' response: We have now improved the clarity of the study, adding following Editorial Suggestion, adding the following sentence at the end of the introduction

Identifying these proteins as early biomarkers could enable risk stratification, guide personalised treatments, and highlight therapeutic targets. Clinically, such biomarkers may inform future tests to identify frail or socioeconomically disadvantaged patients at elevated CVD risk due to exacerbated inflammation and aging.

2. The clinical utility remains unclear

Authors' response: Thank you for your valuable feedback. We respectfully clarify that our paper does not assert immediate clinical applicability but instead seeks to establish a robust foundation for future research. By elucidating the role of inflammatory pathways in frailty, cardiovascular risk, and social determinants of health, our findings aim to guide subsequent validation and longitudinal studies. These are critical steps to translate our insights into clinically actionable strategies, and we appreciate your input in highlighting this important consideration.

3. The daily practice based on study findings remains unanswerable

Authors' response: Thank you for your insightful comment regarding the applicability

of our findings to daily clinical practice. We agree that our study does not yet provide direct guidance for immediate clinical implementation, as this was not its primary objective. Instead, our research aimed to elucidate associations between inflammatory markers, frailty, cardiovascular risk, and social deprivation as a key modifiable factor. These findings lay the groundwork for future validation studies to inform early prevention strategies. At present, we do not advocate specific treatments targeting these markers based solely on our results, and we deeply appreciate your feedback in underscoring this point.